# DIFFERENTIABLE CANONICAL CORRELATION ANALYSIS

**Matthias Dorfer**
Department of Computational Perception
Johannes Kepler University Linz
Linz, 4040, Austria
matthias.dorfer@jku.at

**Jan Schlüter**
The Austrian Research Institute
for Artificial Intelligence
Vienna, 1010, Austria
jan.schlueter@ofai.at

**Gerhard Widmer**
Department of Computational Perception
Johannes Kepler University Linz
Linz, 4040, Austria
gerhard.widmer@jku.at

## ABSTRACT

Canonical Correlation Analysis (CCA) computes maximally-correlated linear projections of two modalities. We propose Differentiable CCA, a formulation of CCA that can be cast as a layer within a multi-view neural network. Unlike Deep CCA, an earlier extension of CCA to nonlinear projections, our formulation enables gradient flow through the computation of the CCA projection matrices, and free choice of the final optimization target. We show the effectiveness of this approach in cross-modality retrieval experiments on two public image-to-text datasets, surpassing both Deep CCA and a multi-view network with freely-learned projections. We assume that Differentiable CCA could be a useful building block for many multi-modality tasks.

## 1 INTRODUCTION

Deep Canonical Correlation Analysis (DCCA) (Andrew et al., 2013) is a non-linear extension of classic Canonical Correlation Analysis (CCA) (Hotelling, 1936) that learns highly correlated latent representations on top of two different neural networks. The central idea of our work is to extend this formulation and cast CCA as a fully differentiable neural network layer which allows for parameter optimization via back-propagation through the CCA projection matrices. This is in contrast to DCCA, where correlation analysis is the topmost part of the network and only used as an optimization target for maximizing the correlation between the respective views. DCCA in general gained a lot of attention recently. It inspired related methods such as Deep Linear Discriminant Analysis (Dorfer et al., 2015) as well as a discriminative re-formulation of DCCA (Elmadany et al., 2016) applied to improve speech-based emotion recognition. Wang et al. (2015a) show that joint optimization of correlation and reconstruction error in auto-encoder configurations is successfully used for representation learning on a multi-modal speech production dataset. We take this as a motivation to evolve and extend the applicability of DCCA.

In our experiments, we employ the proposed differentiable CCA layer in a cross-modality retrieval setup. Cross-modality retrieval is the task of retrieving relevant data of another type when a sample of a different modality is given as a search query. A recent survey by Wang et al. (2016) categorizes the task into *binary* and *real-valued* representation learning. In the case of real-valued representation learning, End-to-End DCCA (Yan & Mikolajczyk, 2015) achieves state of the art retrieval results in combination with retrieval by cosine distance computation. With differentiable CCA, it becomes possible to train the networks to directly minimize the objective which will be used for retrieval (e.g., the cosine distance), while still benefitting from the optimally-correlated projections obtained by CCA. Results on two publicly available datasets (*Flickr30k* (Young et al., 2014), *IAPR TC-12*

(Grubinger et al., 2006)) suggest that our approach is capable to improve retrieval results in both directions.

The remainder of our paper is structured as follows. In Section 2, we review classic and deep CCA, which are the basis for the differentiable CCA layer proposed in Section 3. In Section 4, we show results of an experimental evaluation in a cross-modality retrieval setting and provide further investigations on the representations learned by our networks. Finally, Section 5 concludes the paper.

## 2 FOUNDATION: CANONICAL CORRELATION ANALYSIS AND DEEP CCA

In this section, we review the concepts of classical and deep Canonical Correlation Analysis, the basis for the methodology proposed in this work.

### 2.1 CANONICAL CORRELATION ANALYSIS (CCA)

Let $\mathbf{x} \in \mathbb{R}^{d_x}$ and $\mathbf{y} \in \mathbb{R}^{d_y}$ denote two random vectors with covariances $\Sigma_{xx}$ and $\Sigma_{yy}$ and cross-covariance $\Sigma_{xy}$. The objective of CCA is to find two matrices $\mathbf{A}^* \in \mathbb{R}^{d_x \times k}$ and $\mathbf{B}^* \in \mathbb{R}^{d_y \times k}$ (with $k \le d_x$ and $k \le d_y$) that project $\mathbf{x}$ and $\mathbf{y}$ into a common space maximizing their cross-correlation:

$$(\mathbf{A}^*, \mathbf{B}^*) = \underset{\mathbf{A}, \mathbf{B}}{\arg \max} \ \text{corr}(\mathbf{A}'\mathbf{x}, \mathbf{B}'\mathbf{y}) \tag{1}$$

To fix the scale of $\mathbf{A}$ and $\mathbf{B}$ and force the projected dimensions to be uncorrelated, the optimization is further constrained to $\mathbf{A}'\Sigma_{xx}\mathbf{A} = \mathbf{B}'\Sigma_{yy}\mathbf{B} = \mathbf{I}$, arriving at:

$$(\mathbf{A}^*, \mathbf{B}^*) = \underset{\mathbf{A}'\Sigma_{xx}\mathbf{A}=\mathbf{B}'\Sigma_{yy}\mathbf{B}=\mathbf{I}}{\arg \max} \mathbf{A}'\Sigma_{xy}\mathbf{B} \tag{2}$$

Let $\mathbf{T} = \Sigma_{xx}^{-1/2} \Sigma_{xy} \Sigma_{yy}^{-1/2}$, and let $\mathbf{T} = \mathbf{U}\,\text{diag}(\mathbf{d})\mathbf{V}'$ be the Singular Value Decomposition (SVD) of $\mathbf{T}$ with ordered singular values $d_i \ge d_{i+1}$. As shown by Mardia et al. (1979), we obtain $\mathbf{A}^*$ and $\mathbf{B}^*$ from the top $k$ left- and right-singular vectors of $\mathbf{T}$:

$$\mathbf{A}^* = \Sigma_{xx}^{-1/2}\mathbf{U}_{:k} \qquad \mathbf{B}^* = \Sigma_{yy}^{-1/2}\mathbf{V}_{:k} \tag{3}$$

Moreover, the cross-correlation in the projection space is the sum of the top $k$ singular values:

$$\text{corr}(\mathbf{A}^{*'}\mathbf{x}, \mathbf{B}^{*'}\mathbf{y}) = \sum_{i \le k} d_i \tag{4}$$

In practice, the covariances and cross-covariance of $\mathbf{x}$ and $\mathbf{y}$ are usually not known, but estimated from a training set of $m$ paired vectors, expressed as matrices $\mathbf{X} \in \mathbb{R}^{d_x \times m}, \mathbf{Y} \in \mathbb{R}^{d_y \times m}$:

$$\overline{\mathbf{X}} = \mathbf{X} - \frac{1}{m}\mathbf{X}\mathbf{1} \qquad \overline{\mathbf{Y}} = \mathbf{Y} - \frac{1}{m}\mathbf{Y}\mathbf{1} \tag{5}$$

$$\hat{\Sigma}_{xx} = \frac{1}{m-1}\overline{\mathbf{X}}\overline{\mathbf{X}}' + r\mathbf{I} \qquad \hat{\Sigma}_{xy} = \frac{1}{m-1}\overline{\mathbf{X}}\overline{\mathbf{Y}}' \qquad \hat{\Sigma}_{yy} = \frac{1}{m-1}\overline{\mathbf{Y}}\overline{\mathbf{Y}}' + r\mathbf{I} \tag{6}$$

Here, $r$ is a regularization parameter ensuring the matrices are positive definite. Substituting these estimates for $\Sigma_{xx}$, $\Sigma_{xy}$ and $\Sigma_{yy}$, respectively, we can estimate $\mathbf{A}^*$ and $\mathbf{B}^*$ using Equation 3.

### 2.2 DEEP CANONICAL CORRELATION ANALYSIS (DCCA)

Andrew et al. (2013) propose an extension of CCA that allows learning parametric nonlinear transformations of two variables maximizing the cross-correlation after optimal projection. Specifically, let $\mathbf{a} \in \mathbb{R}^{d_a}$ and $\mathbf{b} \in \mathbb{R}^{d_b}$ denote two random vectors, and let $\mathbf{x} = f(\mathbf{a}; \Theta_f)$ and $\mathbf{y} = g(\mathbf{b}; \Theta_g)$ denote their nonlinear transformations, parameterized by $\Theta_f$ and $\Theta_g$. For example, $f$ and $g$ could be feed-forward neural networks. As before, Equation 3 gives the linear transformations of $\mathbf{x}$ and $\mathbf{y}$ optimizing the CCA objective in Equation 2. Deep CCA optimizes $\Theta_f$ and $\Theta_g$ to further increase the cross-correlation. For $d_x = d_y = k$, the CCA objective is equal to the sum of all singular values of $\mathbf{T}$ (Equation 4), which is equal to its trace norm:

$$\text{corr}(f(\mathbf{a}; \Theta_f), g(\mathbf{b}; \Theta_g)) = \text{corr}(\mathbf{x}, \mathbf{y}) = ||\mathbf{T}||_{\text{tr}} = \text{tr}(\mathbf{T}'\mathbf{T})^{1/2} \tag{7}$$

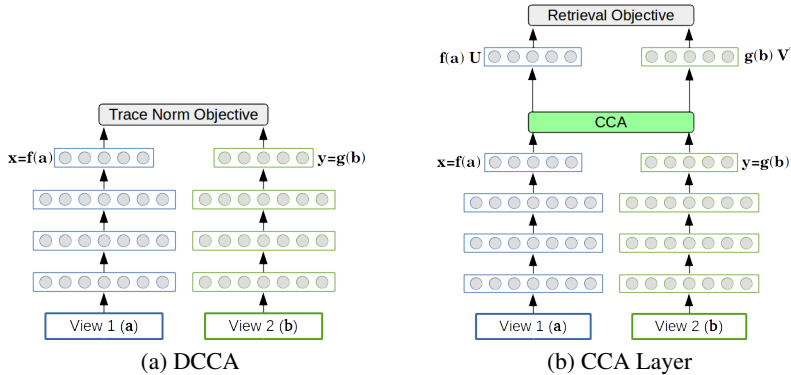

(a) DCCA                           (b) CCA Layer

Figure 1: Comparison of DCCA and the prosed differentiable CCA layer. DCCA optimizes the correlation of the two different views and is therefore the topmost part of the network. In contrast, our CCA layer establishes gradient flow over the CCA computation. This allows us to use the projection output of CCA as input for subsequent components in a multi-view network (e.g., a retrieval objective such as cosine distance).

Andrew et al. (2013) show how to compute the gradient of this *Trace Norm Objective* (TNO) with respect to $\mathbf{x}$ and $\mathbf{y}$. Assuming $f$ and $g$ are differentiable with respect to $\Theta_f$ and $\Theta_g$ (as is the case for neural networks), this allows to optimize the nonlinear transformations via a gradient-based method. Figure 1a shows a schematic sketch of DCCA, as a fixed objective backpropagated through two neural networks.

## 3 DIFFERENTIABLE IMPLEMENTATION OF CCA

In this section, we further extend DCCA to allow not only an arbitrary nonlinear transformation of the inputs, but also arbitrary transformations of (or objectives on) the projected vectors. This allows CCA to be used as a building block within a multi-modality neural network, instead of as a final objective only. In the following, we will discuss how to enable backpropagation through CCA, what to consider when doing stochastic updates, and how to apply it for cross-modality retrieval.

### 3.1 GRADIENT OF CCA

As mentioned above, we can compute the canonical correlation along with the optimal projection matrices from the singular value decomposition $\mathbf{T} = \Sigma_{xx}^{-1/2}\Sigma_{xy}\Sigma_{yy}^{-1/2} = \mathbf{U}\operatorname{diag}(\mathbf{d})\mathbf{V}'$. Specifically, we obtain the correlation as $\sum_i d_i$, and projections as $\mathbf{A}^* = \Sigma_{xx}^{-1/2}\mathbf{U}$ and $\mathbf{B}^* = \Sigma_{yy}^{-1/2}\mathbf{V}$. For DCCA, it suffices to compute the gradient of the total correlation wrt. $\mathbf{x}$ and $\mathbf{y}$ in order to backpropagate it through the two networks $f$ and $g$. Using the chain rule, Andrew et al. (2013) decompose this into the gradients of the total correlation wrt. $\Sigma_{xx}$, $\Sigma_{xy}$ and $\Sigma_{yy}$, and the gradients of those wrt. $\mathbf{x}$ and $\mathbf{y}$. Their derivations of the former make use of the fact that both the gradient of $\sum_i d_i$ wrt. $\mathbf{T}$ and the gradient of $||\mathbf{T}||_{\mathrm{tr}}$ (the trace norm objective in Equation 7) wrt. $\mathbf{T}'\mathbf{T}$ have a simple form; see Andrew et al. (2013, Sec. 7) for details.

For our differentiable CCA, we instead need the gradients of the projected data $\mathbf{A}^{*\prime}\mathbf{x}$ and $\mathbf{B}^{*\prime}\mathbf{y}$ wrt. $\mathbf{x}$ and $\mathbf{y}$, which require $\frac{\partial \mathbf{U}}{\partial \mathbf{x},\mathbf{y}}$ and $\frac{\partial \mathbf{V}}{\partial \mathbf{x},\mathbf{y}}$. We could again decompose this into the gradients wrt. $\mathbf{T}$, the gradients of $\mathbf{T}$ wrt. $\Sigma_{xx}$, $\Sigma_{xy}$ and $\Sigma_{yy}$ and the gradients of those wrt. $\mathbf{x}$ and $\mathbf{y}$. However, while the gradients of $\mathbf{U}$ and $\mathbf{V}$ wrt. $\mathbf{T}$ are known (Papadopoulo & Lourakis, 2000), they involve solving $O((d_x d_y)^2)$ linear $2 \times 2$ systems. To arrive at a more practical implementation that does not require the gradient of the SVD, we reformulate the solution to use two symmetric eigendecompositions $\mathbf{TT}' = \mathbf{U}\operatorname{diag}(\mathbf{e})\mathbf{U}'$ and $\mathbf{T}'\mathbf{T} = \mathbf{V}\operatorname{diag}(\mathbf{e})\mathbf{V}'$ (Petersen & Pedersen, 2012, Eq. 270). This gives us the same left and right eigenvectors we would obtain from the SVD (save for possibly flipped signs, which are easy to fix), along with the squared singular values ($e_i = d_i^2$). The gradients of eigenvectors of symmetric real eigensystems have a simple form (Magnus, 1985, Eq. 7) and both

$\mathbf{TT}'$ and $\mathbf{T}'\mathbf{T}$ are differentiable wrt. $\mathbf{x}$ and $\mathbf{y}$, enabling a sufficiently efficient implementation in a graph-based, auto-differentiating math compiler such as Theano (Theano Development Team, 2016).

## 3.2 STOCHASTIC OPTIMIZATION

For classical CCA, $\Sigma_{xx}$, $\Sigma_{xy}$ and $\Sigma_{yy}$ are estimated from a large set of $m$ training examples (Equation 6). In contrast, gradient-based optimization of neural networks usually estimates the gradients wrt. network parameters from mini-batches of $n$ randomly drawn examples, with $n \ll m$. In Deep CCA as well as in our extension, the correlations are functions of the network parameters that we need to backpropagate through, effectively enforcing $m = n$.

Andrew et al. (2013) solve this discrepancy by optimizing the network parameters with L-BFGS on the full training set, which is infeasible for very large datasets. Yan & Mikolajczyk (2015) instead train on small mini-batches, estimating correlation matrices of size $4096 \times 4096$ from 100 examples only, which seems risky. We will choose a way in between, training on large mini-batches to obtain stable estimates. This approach was also taken by Wang et al. (2015b, Sec. 5.1), who found mini-batches of 400–1000 examples to even outperform full-batch L-BFGS. In addition, for testing, we optionally re-estimate the correlation matrices (and the corresponding projection matrices) using a larger set of $m > n$ examples.

Another tempting option is to train on small mini-batches, but use exponential moving averages updated with each mini-batch as follows:

$$\Sigma_{xx} \leftarrow \Sigma_{xx}(1 - \alpha) + \hat{\Sigma}_{xx}\alpha \quad \Sigma_{xy} \leftarrow \Sigma_{xy}(1 - \alpha) + \hat{\Sigma}_{xy}\alpha \quad \Sigma_{yy} \leftarrow \Sigma_{yy}(1 - \alpha) + \hat{\Sigma}_{yy}\alpha$$

With proper initialization and a sufficiently small coefficient $\alpha$, this gives stable estimates even for small $n$. However, since only the estimates from the current mini-batch $\hat{\Sigma}_{xx}$, $\hat{\Sigma}_{xy}$ and $\hat{\Sigma}_{yy}$ can be practically considered in backpropagation, this changes the learning dynamics: For too small $\alpha$, the projection matrices will be virtually degraded to constants. Empirically, we found that large mini-batches perform slightly better than small batches with moving averages (see Appendix B).

## 3.3 CROSS-MODALITY RETRIEVAL WITH DIFFERENTIABLE CCA

DCCA maximizes the correlation between the latent representations of two different neural networks. When the two network inputs $\mathbf{a}$ and $\mathbf{b}$ represent different views of an entity (e.g., an image and its textual description), DCCA projects them into a common space where they are highly correlated. This can be exploited for cross-modality retrieval: Projecting one modality of an entity, we can find the best-matching representations of the second modality (e.g., an image for a textual description, or vice versa). To find the best matches, a common option is to compute nearest neighbors in terms of cosine distance (Yan & Mikolajczyk, 2015), which is closely related to correlation.

Given the methodology introduced above, we now have the means to optimize DCCA projections directly for the task at hand. In Figure 1b, we show a possible setting where we put the differentiable CCA layer on top of a multi-view network. Instead of optimizing the networks to maximize the correlation of the projected views (the TNO), we can optimize the networks towards a task-specific objective and still benefit from the optimality of the CCA projections.

For this work, we optimize towards minimal cosine distance between the correlated views, the very metric used for retrieval. In the next section, we empirically show that this is indeed beneficial in terms of quantitative retrieval performance as well as convergence speed of network training.

## 4 EXPERIMENTS

We evaluate our approach in cross-modality retrieval experiments on two publicly available datasets (also considered by Yan & Mikolajczyk (2015)) and provide investigations on the representations learned by the network.

## 4.1 EXPERIMENTAL SETUP

For the evaluation of our approach, we consider *Flickr30k* and *IAPR TC-12*, two publicly available datasets for cross-modality retrieval. Flickr30k consists of image-caption pairs, where each image

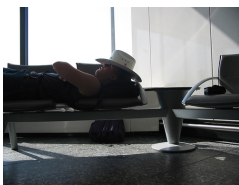

A man in a white cowboy hat reclines in front of a window in an airport.

A young man rests on an airport seat with a cowboy hat over his face.

A woman relaxes on a couch , with a white cowboy hat over her head.

A man is sleeping inside on a bench with his hat over his eyes.

A person is sleeping at an airport with a hat on their head.

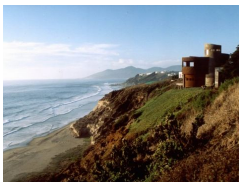

a green and brown embankment with brown houses on the right and a light brown sandy beach at the dark blue sea on the left; a dark mountain range behind it and white clouds in a light blue sky in the background;

Table 1: Example images for Flickr30k (top) and IAPR TC-12 (bottom)

is annotated with five different textual descriptions. The train-validation-test split for Flickr30k is 28000-1000-1000. In terms of evaluation setup, we follow the related work and report results on two different evaluation protocols. Protocol *pooled* pools the five available captions into one "concatenated" text, meaning that only one but richer text annotation remains per image. This is done for all three sets. Protocol *5 captions* pools only the captions of the train set and keeps five separate annotations for validation and test set. The IAPR TC-12 dataset contains 20000 natural images where only one – but compared to Flickr30k more detailed – caption is available for each image. As no predefined train-validation-test split is provided, we randomly select 2000 images for testing, 1000 for validation and keep the rest for training. Yan & Mikolajczyk (2015) also use 2000 images for testing, but did not explicitly mention hold out images for validation. Table 1 shows an example image along with its corresponding captions or caption for either dataset.

The task at hand for both datasets is to retrieve the correct counterpart – either text or image – when given a query element of the other modality. We follow Yan & Mikolajczyk (2015) and use the cosine distance for retrieval in the projection space. As evaluation measures we consider the *Recall@k (R@k)* as well as the *Median Rank (MR)* and the *Mean Average Precision (MAP)*. The *R@k* rate (high is better) is the ratio of queries which have the correct corresponding counterpart in the first $k$ retrieval results. The *MR* is the median position (low is better) of the target in a similarity-ordered list of available candidates. Finally, we define the *MAP* (high is better) as the mean value of $1/Rank$ over all queries.

The input to our networks is a 4096-dimensional image feature vector along with a corresponding text vector representation (5793 for Flickr30k, 2048 for IAPR TC-12). In terms of text pre-processing, we follow Yan & Mikolajczyk (2015), tokenizing and lemmatizing the raw captions as the first step. Based on the lemmatized captions, we compute $l2$-normalized TF/IDF-vectors, omitting words with an overall occurrence smaller than 5 times for Flickr30k and 3 times for IAPR TC-12, respectively. The image represenations are computed from the last hidden layer of a network pretrained on ImageNet (layer *fc7* of *CNN_S* by Chatfield et al. (2014)).

## 4.2 NETWORK ARCHITECTURES AND OPTIMIZATION DETAILS

We feed 4096-dimensional image vectors along with the corresponding text representation into our networks. The image representation is followed by a linear dense layer with 128 units (this will also be the dimensionality $k = 128$ of the resulting CCA retrieval space). The text vector is processed by two batch-normalized (Ioffe & Szegedy, 2015) dense layers of 1024 units each and an ELU activation function (Clevert et al., 2015). As a last layer for the text representation network, we again apply a dense layer with 128 linear units. For a fair comparison, we keep the structure (and number of parameters) of all networks in our experiments the same. The only parameters that vary are the objectives and the corresponding optimization/regularization strategies. In particular, we apply a grid search on the respective hyper-parameters and report the best results for each method. Optimization is performed either using Stochastic Gradient Descent (SGD) with *momentum* or by the *adam* (Kingma & Ba, 2014) update rule.

Table 2: Cross-modality retrieval results on Flickr30k. "E2E-DCCA" is taken from Yan & Mikolajczyk (2015), all other results are our own. Methods marked with "*" re-estimate projection matrices from a larger batch than used during training (10,000 training examples), see Section 3.2.

| Protocol | Method | Image-to-Text | | | | Text-to-Image | | | |
|---|---|---|---|---|---|---|---|---|---|
| | | R@1 | R@5 | R@10 | MR | R@1 | R@5 | R@10 | MR |
| pooled | E2E-DCCA | 27.9 | 56.9 | 68.2 | 4 | 26.8 | 52.9 | 66.9 | 4 |
| | TNO* | 29.9 | 57.9 | 67.9 | 4 | 21.8 | 48.1 | 64.0 | 6 |
| | learned-$cos^2$ | 9.0 | 23.3 | 32.8 | 28 | 8.5 | 23.3 | 32.8 | 26 |
| | CCAL-$l2$ | 18.2 | 42.0 | 53.6 | 9 | 17.7 | 42.2 | 53.2 | 9 |
| | CCAL-$cos$ | 28.9 | 57.5 | 69.1 | 4 | 25.1 | 53.1 | 66.4 | 5 |
| | CCAL-$cos^2$ | 30.7 | 58.8 | 70.1 | 4 | 28.0 | 56.2 | 68.3 | 4 |
| | CCAL-$cos^2$* | 34.1 | 60.0 | 70.6 | 3.5 | 29.2 | 58.3 | 69.7 | 4 |
| 5 captions | E2E-DCCA | 16.7 | 39.3 | 52.9 | 8 | 12.6 | 31.0 | 43.0 | 15 |
| | TNO* | 17.5 | 39.3 | 51.4 | 10 | 13.4 | 31.7 | 41.3 | 19 |
| | CCAL-$cos^2$ | 21.2 | 44.4 | 55.8 | 8 | 14.9 | 35.9 | 47.5 | 12 |
| | CCAL-$cos^2$* | 20.6 | 45.9 | 57.2 | 7 | 15.6 | 37.0 | 49.4 | 11 |

As optimization targets, we consider the following candidates: (1) The Trace Norm Objective (*TNO*) as our base line for cross-modality retrieval (Yan & Mikolajczyk, 2015). (2) The proposed differentiable CCA layer in combination with the objectives cosine distance (*CCAL-cos*), squared cosine distance (*CCAL-cos²*) and euclidean distance (*CCAL-l2*). As an additional setting, we consider a freely-learnable projection layer where the projection matrices $\mathbf{A}$ and $\mathbf{B}$ are randomly initialized weights that can be optimized by the network using SGD in the conventional way. This allows to assess the benefit of using CCA-derived projections within a multi-view network under otherwise unchanged objectives. For this experiment, we optimize for the squared cosine distance and denote the setting by *learned-cos²*. The batch size is set to 1000 samples to allow stable covariance estimates for the CCA (Section 3.2). For further stabilization, we regularize the covariance matrices (Andrew et al., 2013) by adding scaled ($r = 10^{-3}$) identity matrices to the estimates $\Sigma_{xx}$, $\Sigma_{yy}$ and $\mathbf{T}$ (Section 2.1). The variants based on differentiable CCA are additionally regularized by $L2$ weight decay. No dropout is used in this settings as it harmed optimization in our experiments. When optimizing with the TNO we follow Yan & Mikolajczyk (2015) and use dropout ($p = 0.5$) after the first two dense layers of the text network. In Table 4 in Appendix A we provide the optimization settings for all configurations in detail, found using a grid search optimizing MAP on the validation set.

## 4.3 EXPERIMENTAL RESULTS ON CROSS-MODALITY RETRIEVAL

Table 2 lists our results on Flickr30k. Along with our experiments, we also show the results reported in (Yan & Mikolajczyk, 2015) as a reference (E2E-DCCA). However, a direct comparison to our results may not be fair: E2E-DCCA uses a different ImageNet-pretrained network for the image representation, and finetunes this network while we keep it fixed (as we are only interested in comparing differentiable CCA to alternatives, not in obtaining the best possible results). Our TNO results use the same objective as E2E-DCCA, but our network architecture, permitting direct comparison.

When comparing the performance of our networks, we observe a gain both for image-to-text and text-to-image retrieval when training with the CCAL-$cos^2$ objective compared to TNO (e.g., R@1 of 34.1 compared to 29.9 under protocol *pooled*). This indicates that training a network directly on the objective used for retrieval (using differentiable CCA) is a reasonable design choice. A closer look at the results also reveals that the squared cosine distance is superior compared to the remaining objectives. We further observe that the randomly initialized projection matrices learned entirely by SGD (learned-$cos^2$) show poor performance compared to their CCA counterpart (even though in theory, they could converge to exactly the same solution). This suggests that exploiting the beneficial properties of the CCA projections directly within a network during training is a powerful tool, supporting optimization of related objectives. CCAL-$l2$ for example performs poorer than the variants including cosine losses but still better than the version with learned weights. On protocol

Table 3: Cross-modality retrieval results on IAPR TC-12

| Method | Image-to-Text | | | | Text-to-Image | | | |
|---|---|---|---|---|---|---|---|---|
| | R@1 | R@5 | MAP | MR | R@1 | R@5 | MAP | MR |
| E2E-DCCA | 30.2 | 57.0 | 0.426 | | 29.5 | 60.0 | 0.415 | |
| TNO* | 30.0 | 56.7 | 0.424 | 4 | 28.0 | 55.4 | 0.410 | 5 |
| CCAL-$cos^2$* | 31.1 | 58.4 | 0.439 | 4 | 26.8 | 55.1 | 0.403 | 4 |

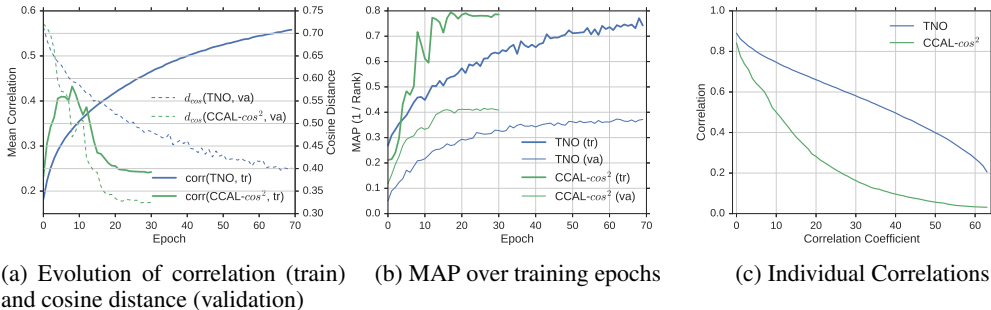

(a) Evolution of correlation (train) and cosine distance (validation)

(b) MAP over training epochs

(c) Individual Correlations

Figure 2: Comparison of the TNO and CCAL-$cos^2$ based on the total amount of canonical correlation (sum over singular values $\mathbf{d}$) as well as the cosine distance between corresponding samples.

*5 captions*, we only report the best results (CCAL-$cos^2$) along with the TNO and observe similar tendencies. Note that there are various other methods reporting results on Flickr30k (Karpathy et al., 2014; Socher et al., 2014; Mao et al., 2014; Kiros et al., 2014) which partly surpass ours, for example by using more elaborate processing of the textual descriptions. We omit these results as we focus on the comparison of DCCA with the proposed differentiable CCA layer.

In Table 3, we list our results on the IAPR TC-12 dataset. We again show the retrieval performances of Yan & Mikolajczyk (2015) as a baseline (again with limited comparability, due to a different architecture and a different train-validation-test split), along with our implementation of the TNO and the CCA layer trained with squared cosine distance. For image-to-text retrieval, we achieve slightly better retrieval performances when training with cosine distance and propagating the gradients back through the differentiable CCA layer. For the other direction, results are slightly worse.

## 4.4 INVESTIGATIONS ON LEARNED REPRESENTATIONS

In this section, we provide a more detailed look at the learned representations. We compare the representations learned with the TNO to the proposed CCA layer optimized with the squared cosine distance objective. For easier comparison, we re-train both networks with a reduced projection dimensionality of $h = 64$ – otherwise, the TNO takes much longer to converge than the CCA layer. This results in slightly decreased performance for both, but the relative tendencies are preserved.

Figure 2a shows the evolution of the mean correlation (mean over singular values with maximum 1.0) on the training set during optimization. Allong with the correlation, we also plot the average cosine distance between corresponding pairs on the validation set. As expected, for the TNO we observe a continous decrease of cosine distance when the correlation increases. Interestingly, this is not the case for CCAL-$cos^2$. The result suggests that the network found a way of minimizing the cosine distance other than by increasing correlation between the representations – the latter even decreases after a few training epochs. In Figure 2b, we plot the corresponding evolution of MAP on the training and validation set, confirming that the decreased cosine distance indeed also leads to improved retrieval performance. Finally, in Figure 2c we compare the individual correlation coefficients (magnitudes of CCA singular values on the training set) of both representations after the last training epoch. This details the observation in Figure 2a: not only the total correlation, but also the individual correlation coefficients are considerably higher when training with TNO, even though the retrieval performance is lower.

## 5 CONCLUSION

We presented a fully differentiable version of Canonical Correlation Analysis which enables us to back-propagate errors directly through the computation of CCA. As this requires to establish gradient flow through CCA, we formulate it to allow easy computation of the partial derivatives $\frac{\partial \mathbf{A}^*}{\partial \mathbf{x},\mathbf{y}}$ and $\frac{\partial \mathbf{B}^*}{\partial \mathbf{x},\mathbf{y}}$ of CCA's projection matrices $\mathbf{A}^*$ and $\mathbf{B}^*$ with respect to the input data $\mathbf{x}$ and $\mathbf{y}$. With this formulation, we can incorporate CCA as a building block within multi-modality neural networks that produces maximally-correlated projections of its inputs. In our experiments, we use this building block within a cross-modality retrieval setting, optimizing a network to minimize the cosine distance of the correlated CCA projections. Experimental results show that when using the cosine distance for retrieval (as is common for correlated views), this is superior to optimizing a network for maximally-correlated projections (as done in Deep CCA), or not using CCA at all. We further observed (Section 4.4) that it is not necessarily required to have maximum correlation to achieve a high retrieval performance. Finally, our differentiable CCA layer could provide a useful basis for further research, e.g., as an intermediate processing step for learning binary cross-modality retrieval representations.

ACKNOWLEDGMENTS

The research reported in this paper has been supported by the Austrian Federal Ministry for Transport, Innovation and Technology, the Federal Ministry of Science, Research and Economy, and the Province of Upper Austria in the frame of the COMET center SCCH, as well as by the Federal Ministry for Transport, Innovation & Technology (BMVIT) and the Austrian Science Fund (FWF): TRP 307-N23. The Tesla K40 used for this research was donated by the NVIDIA Corporation.

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

## APPENDIX A: OPTIMIZATION SETTINGS

The table below provides a detailed listing of the optimization strategies for all our experiments. All our configurations are of course also available in our experimental code published at (will be added).

### Flickr30k

| Objective | Optimizer | Units | $lr_{ini}$ | $lr$-schedule | Dropout | $L2$ | $r$ |
|---|---|---|---|---|---|---|---|
| TNO | momentum | 2048 | 0.05 | constant | 0.5 | none | $10^{-3}$ |
| CCAL | momentum | 1024 | 0.5 | $\times 0.7$ from epoch 10 | none | 0.002 | $10^{-3}$ |
| learned-$cos^2$ | momentum | 1024 | 0.25 | none | none | 0.002 | $10^{-3}$ |

### IAPR TC-12

| Objective | Optimizer | Units | $lr_{ini}$ | $lr$-schedule | Dropout | $L2$ | $r$ |
|---|---|---|---|---|---|---|---|
| TNO | adam | 1024 | 0.001 | $\times 0.1$ in epoch 30 | none | 0.0001 | $10^{-3}$ |
| CCAL | adam | 1024 | 0.001 | $\times 0.1$ in epoch 50 | none | 0.0002 | $10^{-3}$ |

Table 4: Details on optimization strategies for the respective networks

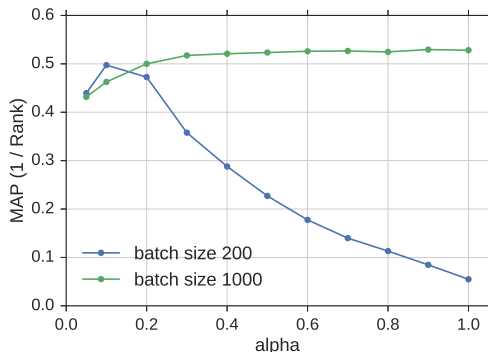

Figure 3: Influence of parameter $\alpha$.

## APPENDIX B: INFLUENCE OF RUNNING AVERAGE STATISTICS

In this additional section, we investigate the influence of weighting coefficient $\alpha$ when using exponential moving average estimates of the covariance matrices for CCA computation (see Section 3). A high $\alpha$ (close to 1.0) means that the averaged estimate of $\Sigma_{xx}$, $\Sigma_{yy}$ and $\Sigma_{xy}$ mostly depends on the current batch, and a low $\alpha$ (close to 0.0) means it more strongly depends on the history of previous batches. To assess whether and under what circumstances exponential moving averages are helpful, we run an additional experiment on the IAPR TC-12 dataset as follows: We re-train one of the models of Section 4 both with batch size 1000 and with batch size 200, varying $\alpha$ from 1.0 to 0.1 with a step size of 0.1 and measuring the *MAP* achieved on the validation set. We run each setting three times and report the average over the three runs. Figure 3 shows the results of this experiment. For batch size 1000, we draw the same conclusion as was reported in (Wang et al., 2015a;b): If the batch size is sufficiently large and representative for the entire population, learning on distribution parameters (in this case covariance matrices) is feasible, and the network performs best when trained with an $\alpha$ close to one. This is not the case for batch size 200. In particular, the configurations with a large $\alpha$ (small effective running average window) perform poorly. We conclude that a batch size of 200 is too small to obtain stable and representative covariances. However, when choosing a small $\alpha$, it is still possible to train the models and achieve reasonable retrieval performance. As a practical recommendation, we suggest to use large batch sizes whenever possible (e.g., if feasible with available hardware). If the batch size needs to be reduced (e.g., for very large models and limited memory), using small alpha values still allows for training canonically correlated retrieval networks. For this work, we use a batch size of 1000 and fix $\alpha = 1$, disabling moving averages.

