# Peer review of "Differentiable Canonical Correlation Analysis"

_ICLR 2017 — rejected_

[Official Review · AnonReviewer2 · rating 3 · confidence 4 · 15 Dec 2016]
**Needs significant work before it can be publishable**

The authors propose to combine a CCA objective with a downstream loss.  This is a really nice and natural idea.  However, both the execution and presentation leave a lot to be desired in the current version of the paper.

It is not clear what the overall objective is.  This was asked in a pre-review question but the answer did not fully clarify it for me.  Is it the sum of the CCA objective and the final (top-layer) objective, including the CCA constraints?  Is there some interpolation of the two objectives?  

By saying that the top-layer objective is "cosine distance" or "squared cosine distance", do you really mean you are just minimizing this distance between the matched pairs in the two views?  If so, then of course that does not work out of the box without the intervening CCA layer:  You could minimize it by setting all of the projections to a single point.  A better comparison would be against a contrastive loss like the Hermann & Blunsom one mentioned in the reviewer question, which aims to both minimize the distance for matched pairs and separate mismatched ones (where "mismatched" ones can be uniformly drawn, or picked in some cleverer way).  But other discriminative top-layer objectives that are tailored to a downstream task could make sense.

There is some loose terminology in the paper.  The authors refer to the "correlation" and "cross-correlation" between two vectors.  "Correlation" normally applies to scalars, so you need to define what you mean here.  "Cross-correlation" typically refers to time series.  In eq. (2) you are taking the max of a matrix.  Finally I am not too sure in what way this approach is "fully differentiable" while regular CCA is not -- perhaps it is worth revisiting this term as well.

Also just a small note about the relationship between cosine distance and correlation:  they are related when we view the dimensions of each of the two vectors as samples of a single random variable.  In that case the cosine distance of the (mean-normalized) vectors is the same as the correlation between the two corresponding random variables.  In CCA we are viewing each dimension of the vectors as its own random variable.  So I fear the claim about cosine distance and correlation is a bit of a red herring here.

A couple of typos:

"prosed" --> "proposed"
"allong" --> "along"

[Official Review · AnonReviewer3 · rating 4 · confidence 4 · 16 Dec 2016]
**paper needs to be more explicit**

After a second look of the paper, I am still confused what the authors are trying to achieve.

The CCA objective is not differentiable in the sense that the sum of singular values (trace norm) of T is not differentiable. It appears to me (from the title, and section 3), the authors are trying to solve this problem. However,

-- Did the authors simply reformulate the CCA objective or change the objective? The authors need to be explicit here.

-- What is the relationship between the retrieval objective and the "CCA layer"? I could imagine different ways of combining them, such as combination or bi-level optimization. And I could not find discussion about this in section 3. For this, equations would be helpful.

-- Even though the CCA objective is not differentiable in the above sense, it has not caused major problem for training (e.g., in principle we need batch training, but empirically using large minibatches works fine). The authors need to justify why the original gradient computation is problematic for what the authors are trying to achieve. From the authors' response to my question 2, it seems they still use SVD of T, so I am not sure if the proposed method has advantage in computational efficiency.

In terms of paper organization, it is better to describe the retrieval objective earlier than in the experiments. And I still encourage the authors to conduct the comparison with contrastive loss that I mentioned in my previous comments.

[Official Review · AnonReviewer1 · rating 3 · confidence 4 · 21 Dec 2016]
**Unclear about the contribution**

It is not clear to me at all what this paper is contributing. Deep CCA (Andrew et al, 2013) already gives the gradient derivation of the correlation objective with respect to the network outputs which are then back-propagated to update the network weights. Again, the paper gives the gradient of the correlation (i.e. the CCA objective) w.r.t. the network outputs, so it is confusing to me when authors say that their differentiable version enables them to back-propagate directly through the computation of CCA.

[Author Response · Matthias Dorfer · 04 Jan 2017]
**Response to reviewer comments**

Dear reviewers, thank you for your effort and valuable feedback! We 
understand that we did not manage to clearly present the central idea of 
our work.
We do *not* combine the CCA objective (i.e., maximizing the correlation 
between the two hidden representations, the so-called Trace Norm 
Objective in Deep CCA) with another objective. We use CCA as a 
transformation anywhere within a network that optimally correlates two 
views, but we do not optimize the network towards maximal correlation as 
done in Deep CCA. Instead, we apply the CCA transformations in the 
forward pass (not done in Deep CCA) and compute an arbitrary loss 
function on the transformed data.
Our use of CCA can be compared to Batch Normalization (BN): BN provides 
a transformation applicable anywhere within a network that normalizes a 
single view, and allows backpropagation of gradients through the 
normalization procedure. Similarly, we backpropagate gradients through 
the CCA procedure, where the gradients can be derived from any loss 
function operating on the optimally-correlated views produced by CCA.
We are grateful for all your comments, and will rewrite our manuscript 
to submit it elsewhere.

[Final Decision · Program Chairs · 06 Feb 2017]
**ICLR committee final decision**

The authors propose to use CCA as a transformation within a network that optimally correlates two views. The authors then back-propagate gradients through the CCA. Promising experimental results on for cross-modality retrieval experiments on two public image-to-text datasets are presented. 
 
 The main concern with the paper is the clarity of the exposition. The novelty and motivation of the approach remains unclear, despite significant effort from the reviewers to understand. 
 
 A major rewriting of the paper will generate a stronger submission to a future venue.